# Microstructure and Wear Resistance of Laser Cladding of Fe-Based Alloy Coatings in Different Areas of Cladding Layer

**DOI:** 10.3390/ma14112839

**Published:** 2021-05-26

**Authors:** Qiaofeng Bai, Changyao Ouyang, Chunjiang Zhao, Binhui Han, Yingliang Liu

**Affiliations:** 1School of Mechanical Engineering, Taiyuan University of Science and Technology, Taiyuan 030024, China; 2013001@tyust.edu.cn (Q.B.); S20190160@stu.tyust.edu.cn (C.O.); 2School of Aviation Maintenance Engineering, Xi’an Aviation Vocational and Technical College, Xi’an 710089, China; 2004059@tyust.edu.cn; 3Shuozhou Jinhua Industrial Co., Ltd., Shuozhou 036000, China; S20190163@stu.tyust.edu.cn

**Keywords:** laser cladding, Fe-based alloy coating, microstructure, hardness, wear resistance

## Abstract

In this study, laser cladding technology was used to prepare Fe-based alloy coating on a 27SiMn hydraulic support, and a turning treatment was used to obtain samples of the upper and middle regions of the cladding layer. The influence of microstructure, phase composition, hardness, and wear resistance in different areas of the cladding layer was studied through scanning electron microscopy (SEM), X-ray diffractometry (XRD), friction and wear tests, and microhardness. The results show that the bcc phase content in the upper region of the cladding layer is less than that in the middle region of the cladding layer, and the upper region of the cladding layer contains more metal compounds. The hardness of the middle region of the cladding layer is higher than that of the upper region of the cladding layer. At the same time, the main wear mechanism of the upper region of the cladding layer is adhesive wear and abrasive wear. The wear mechanism of the middle region of the cladding layer is mainly abrasive wear, with better wear resistance than the upper region of the cladding layer.

## 1. Introduction

A 27SiMn hydraulic support is an important part of mining hydraulic machinery, which operates in a complex environment. To ensure the safety of workers and to increase the service life of the machine, its surface can be modified to improve the wear resistance of the pillar and various material properties. Compared with the commonly used thermal spraying and chemical bonding techniques used for preparing wear-resistant coatings, laser cladding technology uses high-energy laser beams to melt and weld metal powder on the surface of the substrate, so that the coating and the substrate have high metallurgical properties. Combined with a lower dilution rate, rapid cooling of the coating affects the refinement of the structure and changes the crystal grain and, therefore, is beneficial in improving the wear resistance of the coating.

In recent years, laser cladding has been increasingly used in additive manufacturing as a green manufacturing technology; the equipment, technology, materials, and industrial applications of laser cladding are all involved [1].

Extensive research has been conducted on laser cladding equipment, processes, materials, and applications of laser cladding technology in industry. Jiali Zhou et al. [2] used laser cladding technology to prepare Ti_3_AlC_2_ reinforced cobalt-based alloy coating on the surface of H13 steel and studied the friction and wear properties and oxidation behavior of the coating. The results showed that, as the mass fraction of Ti_3_AlC_2_ increased, it exhibited a good friction reduction effect. The O content of the Co-based alloy coating increased with an increase in Ti_3_AlC_2_ mass fraction. At 600 °C, the O content was less than 3%, which had good antioxidant properties. X. P. Tao et al. [3] used laser cladding technology to prepare aluminum bronze coatings with different iron and nickel contents on the surface of 316 stainless steel and studied the influence of different iron and nickel contents on their wear resistance. Li Jiahong et al. [4] used laser cladding technology to prepare Cr–Ni coatings with different mass ratios at 600° on H13 hot processed model steel and studied the friction and wear properties of the Cr–Ni coatings. The results showed that the average friction coefficient of the 20% Cr–80% Ni coating was the lowest and that it showed better anti-friction performance. A Spanish scholar, Juan Pereira [5], studied the dry friction and wear properties of NiCoCrAlY and CoNiCrAlY laser cladding layers and AISI (Martensitic stainless steel) 304 austenitic stainless steel at low and high temperatures. Xu Huang [6] studied the effects of laser power, scanning speed, gas powder flow rate, and TiC mass fraction on the phase composition, microstructure, and element distribution of Ni35A/TiC cladding layer and identified the cause of the wear behavior. The results provided valuable support for the microstructure control and wear behavior prediction of the composite cladding layer. Bo Sun et al. [7] studied the coating quality, organization, and wear properties of FeBSiNb coatings using wide-beam laser cladding. To improve the hardness and wear resistance of cold-working molds, Jiangzheng Shi et al. [8] used laser cladding technology to prepare Al_2_O_3_ reinforced WC-10Co4Cr coating on the surface of Cr_12_MoV steel. The results showed that the addition of Al_2_O_3_ was the main factor for reducing the COF (coefficients of friction) of the coating and improving the wear resistance.

At present, processing methods such as turning, grinding, and polishing are mainly used to make the cladding layer reach the required thickness and roughness after laser cladding [9]. There are, however, a few studies on the thickness and cladding areas of the cladding layer. In this paper, 27SiMn steel is used as the base material for laser cladding of Fe-based coating. The microstructure, microhardness, friction, and wear characteristics of different regions were studied using a turning treatment to process samples of the upper and middle regions of the cladding layer.

## 2. Materials and Methods

### 2.1. Material

The substrate used was 27SiMn (Φ150 × 60 mm in size). Before cladding, sandpaper was used to remove rust and oil stains until the surface showed metallic luster, and then it was cleaned with ethanol. The cladding powder was Fe-based alloy powder (average hardness 51 HRC), with an average diameter of 38 μm and spherical shape that ensured better flow properties. It was composed of Fe, Cr, Ni, and Si powders, as well as other trace element powders. Its microscopic appearance is shown in Figure 1, and its chemical composition is listed in Table 1.

### 2.2. Laser Cladding Process

Model RFL-C4000 (Raycus Fiber Laser, Wuhan, China) high-power fiber optics, Model DPSF-2 (Dual Package System Framework, Shanghai, China) powder feeding (the protective gas and powder feed gas are both Ar), and synchronous powder feed laser cladding were performed on the substrate. The process parameters were laser power P = 2.7 kW, powder feeding speed V1 = 2.8 rad/min, cladding rate V2 = 9 m/min, defocus 15 mm, and overlap rate = 80%. The cladding test was carried out with this combination of process parameters, and then the surface of the obtained cladding layer was treated with a lathe. The initial cladding layer was processed in different areas, named Sample A (the upper region of the cladding layer, thickness was about 1.1 mm) and Sample B (the middle region of the cladding layer, thickness was about 0.55 mm). Figure 2 illustrates the schematic diagram of laser cladding.

### 2.3. Performance Testing

The cladding layer was cut by wire cutting into small metal samples (11 × 11 × 11 mm^3^) that contained the cladding layer. The cross-sections of the cladding layer of the metal samples were ground and polished, and ferric chloride was used to corrode the samples. A Zeiss (ZEISS) field emission scanning electron microscope ∑IGMA 300 (SEM) (Carl Zeiss, Jena, Germany) was used to observe the microstructure and morphology of the samples of different areas of the cladding layer. An Empyrean X-ray diffractometer (PANalytical, Almelo, The Netherlands) was used to determine the phase of the samples. The test conditions were Cu target, the diffraction range was 20°–80°, and the diffraction speed was 2°/min. The phase composition analyses of the samples were conducted in different areas of the cladding layer. The microhardness of the cladding layer section of each sample and the substrate was measured using an HXD-1000TM digital microhardness tester (Changfang, Shanghai, China). Microhardness was measured at intervals of 50 μm along with the depth of the layer, with a loading load of 300 g and a holding time of 15 s. Finally, a multifunctional friction and wear tester (CFT-I, ZhongkeKaihua, Lanzhou, China) was used to conduct multiple dry friction and wear tests on the cladding layer samples of different cladding areas. The diameter of the friction pair was 5 mm for the Si3N4 ceramic ball, the normal load was 30 N, the speed was 500 r/min, the stroke was 5 mm, and the time was 30 min. An electronic analytical balance was used (the balance had a range of 220 mg and an accuracy of 0.1 mg). The quality of the samples and grinding balls was recorded before and after each test.

## 3. Results and Discussion

### 3.1. Macromorphology of the Cladding Layer

Figure 3 shows the molten pool morphology and turning diagrams after laser cladding and the turning surface treatment. It can be seen that the size of the cladding layer is relatively uniform. There is no obvious trough-like sinking phenomenon between two adjacent tracks, as well as no cracks on the surface or air bubbles. The surface of the cladding layer is smooth after turning in different cladding areas, and there are no cracks, bubbles, or pits of large particles. Figure 4 shows the cladding layer flaw detection and its cross-sectional metallographic microscope observation and cross-sectional schematic diagram. A colored flaw detection agent was used for flaw detection on the surface of the cladding layer, and the test showed that there was no evidence of crack phenomenon. The 11 mm cube was obtained by wire cutting, and the section was ground and polished with high-granularity sandpaper. A metallurgical microscope was used to observe the whole section of the cladding layer, and no cracks or pores were observed. There is a clear white strip with no inclusions between the cladding layer and the substrate, indicating that the cladding layer and the substrate exhibit a good metallurgical bond.

### 3.2. Phase Analysis

The XRD spectra of Samples A and B are shown in Figure 5. It can be observed from the figure that the phase of Sample A (the upper region of the cladding layer) is mainly composed of the body-centered cubic (bcc structure) phase, intermetallic compounds M_5_C_2_, and a small amount of face-centered cubic (fcc structure) phase. The phase of Sample B (the middle region of the cladding layer) is mainly composed of the bcc phase and a small amount of the fcc phase. The bcc phase diffraction intensity of Sample A’s cladding layer is 11,821, and the bcc phase diffraction intensity of Sample B’s cladding layer is 16,050. Compared with Sample B, the bcc phase diffraction intensity of Sample A’s cladding layer is weaker, indicating that the bcc phase content of Sample A’s cladding layer is less than that of sample B’s cladding layer. The cladding layer of Sample A contains more metal compounds.

### 3.3. Microstructure Analysis

Figure 6 shows the morphology of the different areas of the cladding layer. The microstructure of Sample A includes the lower part of the cladding layer, the middle part of the cladding layer, and the upper region of the cladding layer; the microstructure of Sample B includes the lower part of the cladding layer and the middle area of the cladding layer.

Figure 6a,d shows the microstructure of the lower area of the cladding layer. The interface between the cladding layer and the substrate is smooth, and no pores, cracks, or slag inclusions are observed at the junction of the two. The lower part of the cladding layer is mostly flat crystals and granular or short rod-shaped cell crystals. On the flat and cell crystals, there are coarse columnar crystal structures that grow in various directions, perpendicular to the interface. The formation of flat crystals is mainly due to the low crystallization rate, R, at the substrate interface of the molten pool. Here, the temperature gradient, G, is relatively large, the G/R value is large, and no component overcooling occurs. At this time, all the heat released by solidification dissipates in the solid on the surface of the interface, causing the crystal surface to slowly move forward; therefore, the crystal is in a flat state. With an increase in the cladding thickness, the G/R value gradually decreases and component undercooling occurs, the planar crystal begins to turn to cellular crystals, and coarse columnar crystals begin to grow [10].

Figure 6b,e shows the microstructure of the middle region of the cladding layer. It can be observed that the microstructure in the middle region of the cladding layer is mainly composed of fishbone-like dendrites and elongated dendrites with finer grain sizes, because, in the middle region of the cladding layer, the solidification rate of the molten pool becomes higher, and the supercooling zone of the composition at the front of the liquid–solid interface gradually increases. The increase in length of the dendrite is much greater than the increase in width. The middle region of the cladding layer is the transition area of the cladding layer, connecting the lower and upper regions. The heat dissipation in the molten pool mainly depends on the reverse discharge of the matrix, and the molten pool has the characteristics of directional solidification; therefore, the increase in the length of dendrites is much greater than the increase in width [11]. The dendrites in the cladding layer mostly have different growth directions, which are mainly caused by the inconsistent heat source in the cladding layer and the different heat dissipation directions. Related conclusions have also been obtained by Wang Rongjian et al. [12].

Figure 6c shows the microstructure of the upper region of the cladding layer. In the upper region of the cladding layer, there are mainly dendrites, long dendrites to short dendrites, and equiaxed crystals with smaller grain sizes. In the surface layer, the cladding layer is smaller, because the surface layer is in contact with the outside air and the heat dissipation rate is fast. During the growth process of a dendrite, its branches are disconnected and freed, and the dendrites produced in the freeing process are broken to form equiaxed crystals [13].

Figure 7 shows the diffusion diagram of the elements at the interface between the cladding layer and the substrate. From the cladding layer to the substrate, the Fe element content increases, and the Cr element content decreases, mainly because the 27SiMn substrate has higher iron content than the cladding layer and the cladding layer has a higher Cr content than the substrate.

Figure 8 is a scanning map of the upper and middle regions of the cladding layer. It can be seen that Cr, Si, and Ni elements segregate more obviously in the upper region of the cladding layer than in the middle region of the cladding layer. This is due to the segregation of crystals caused by the selective crystallization in the metal solidification process. As a result of the excessively fast cooling rate of laser cladding, the structure of the cladding layer solidifies and crystallizes faster. During the solidification process, because the middle and lower regions of the crystalline accumulation layer shrink and sink, and the upper region cannot sink at the same time, micro-cracks are generated on the accumulation layer. The cracks are filled with low-melting solutes, and segregation is formed. The C element in the upper region of the cladding layer is segregated, and it is mainly carbides here.

### 3.4. Hardness Testing

Figure 9 shows the microhardness of the cladding layers of Samples A and B. It can be observed from Figure 9 that the hardness values of the cladding layers of Samples A and B are higher than that of the substrate and distributed in a stepped manner. The hardness in the cladding area is much higher than that of the matrix. Approaching the heat-affected zone, the hardness begins to decrease, and the hardness in the heat-affected zone gradually decreases to the hardness of the matrix. In the cladding zone, this is mainly due to the hard phases such as borides and carbides crystallized. In the heat-affected zone, the liquid metal rapidly solidifies, and base remelting occurs. The mixing degrees of the cladding material and the base material are different, resulting in an alloy composition representing the difference between them; the grains are coarse, and the microhardness is significantly reduced [14].

It can be observed from the figure that the average hardness value of the middle region of Sample A is 476.6 HV_0_._3_, which is larger than the average hardness value of the upper region, and the hardness value of the upper region of Sample B is 500.1 HV_0_._3_, which reaches the maximum. It can be seen that the hardness of the middle region of the cladding layer is higher than that of the upper region of the cladding layer, significantly higher than that on the surface. This is because, on the surface of the cladding layer, the action of the high-energy-density laser beam causes the volatilization and burning of alloy elements, resulting in the appearance of fine cavities in the surface structure. Although the surface grains of the cladding layer are relatively small, the hardness is still low. The dendrites in the middle region of the cladding layer hinder the movement of dislocations and are also the main reason for the increase in hardness. GeYaqiong et al. [15] and Zhang Kaiyi et al. [16] also independently reached similar conclusions and reported that the central region had higher hardness. At the same time, there is more bcc phase in the middle region of the cladding layer, and the lattice resistance of dislocation movement in the bcc phase is higher, which makes the middle region stronger. The ability to resist plastic deformation is enhanced, and plastic deformation becomes difficult, which makes this region high in hardness.

### 3.5. Abrasion Resistance Test

The multifunctional friction and wear tester tests the wear resistance. The friction and wear test parameters are shown in Table 2.

Figure 10 shows the wear and friction coefficient curve diagram of the cladding layer and the substrate. It can be observed from Figure 10a that the average mass loss of Sample A on the surface of the cladding layer and Sample B in the middle region of the cladding layer is 0.8 and 0.3 mg, respectively; the average mass loss of the 27SiMn matrix is 1.2 mg. Relative wear resistance ε is calculated as follows [17]:(1)$$\varepsilon =\frac{{\Delta m}_{0}}{\Delta m}$$
where Δm_0_ is the amount of wear of the matrix sample (mg) and Δm is the amount of wear of the cladding layer sample (mg).

The relative wear resistances of Samples A and B are 1.5 and 4, respectively. The analysis shows that the wear resistance of the sample located in the middle region of the cladding layer is about 2.7 times higher than that of the sample located on the surface of the cladding layer.

Figure 10b shows the friction coefficient curve between the cladding layer and the substrate. The average friction coefficients of Samples A and B and the substrate are 0.55, 0.36, and 0.59, respectively. The average friction coefficient of the cladding layer is lower than that of the substrate, and the average friction coefficient of Sample B is also lower than that of Sample A. In the initial stage of wear, the friction coefficient of the cladding layer and the substrate fluctuates significantly, because, in the initial stage of wear, the contact area between the sample and the grinding ball is small, the contact stress is large, and the friction coefficient increases rapidly. After that, the contact stress continues to decrease, and the friction coefficient tends to stabilize [18,19]. In the stable period of running-in, the friction coefficient of Sample B fluctuates less than that of Sample A, and the curve is more stable.

Figure 11 shows the wear profile of the cladding layer and the substrate. It can be observed that the wear scar width and depth of the substrate are the largest. The width and depth of the wear scar of Sample B are the smallest. It shows that the wear resistance of the cladding layer is better than that of the matrix, and the wear resistance of Sample B is better than that of Sample A. It shows that the middle region of the cladding layer is more wear-resistant than the upper region of the cladding layer.

### 3.6. Wear Mechanism

Figure 12 shows the wear morphology of Samples A and B. Figure 12a,b shows that there are a lot of furrows with different lengths and widths on the wear surface of Sample A, the direction is parallel to the sliding direction of the ball, and there is a concave structure on the wear scar surface. The scratch depth on the surface of the sample is more obvious, and the spalling pit is also more obvious. It can be seen that Sample A has undergone serious plastic deformation after abrasion. It shows that the main wear mechanisms are adhesive wear and abrasive wear [20,21,22,23]. Kuang et al. [20] also reported similar conclusions. Figure 12c,d shows that after friction and wear of Sample B, fine abrasive grains and shallow and dense micro-plows with parallel directions appear on the wear surface. No obvious sticking or peeling phenomena are seen. The bottom is relatively flat, and only a slight degree of wear has occurred. The wear mechanism is mainly abrasive wear. It can be seen that the wear resistance of Sample B is better than that of Sample A, and the wear resistance of the middle region of the cladding layer is better than that of the upper region of the cladding layer.

This is because when the upper region of the cladding layer is finally solidified, due to the shrinkage of the volume, concentrated shrinkage holes or scattered looseness is formed in it, and the surface layer is in contact with more external impurities [24], making the surface organization loose, with small cavities and gaps. The central region of the cladding layer is mainly composed of fishbone-like and elongated dendrites with finer grain sizes. The dendrites adhere to the upper and lower regions of the cladding layer; therefore, the regional organization has better bonding strength and density. A good wear-resistant framework perpendicular to the growth direction of thin rod dendrites is formed, and its wear resistance is improved.

At the same time, there is an uneven distribution of hard phase carbides in the upper region of the cladding layer, causing stress concentration around it. In the process of continuous friction, the hard phase and the nearby alloy structure are constantly being squeezed. Due to pressure and molecular bonding force, the hard points are torn and peeled off, and as a result, adhesive wear occurs [25,26,27]. At the same time, during reciprocating friction, the fallen hard phase of wear debris can easily form three-body wear with the grinding ball and the coating, which intensifies the wear process. Due to the shedding of a large amount of hard phase, the resistance of Si3N4 to the grinding ball and hard phase is weakened, and the material loss is serious, so the wear is relatively large. Li Keyao et al. [28] and Ouyang Chunsheng et al. [29] also reached similar conclusions.

## 4. Conclusions

Laser cladding technology was used to prepare Fe-based alloy coating on the surface of 27SiMn steel. A turning treatment was used to turn the initial cladding layer in different areas of the cladding layer, i.e., Sample A (the upper area of the cladding layer, the thickness is about 1.1 mm) and Sample B (the middle area of the cladding layer, the thickness is about 0.55 mm). The phase, structure, microhardness, and wear resistance of Samples A and B were systematically studied. The main conclusions of this study are as follows:

(1) The phases of the cladding layers of Samples A and B are mainly composed of bcc phase, intermetallic compound, and a small amount of fcc phase. The content of the bcc phase of Sample A’s cladding layer is less than that of Sample B’s cladding layer, but Sample A’s cladding layer contains more metal compounds and carbide phases. The surface structure of Sample A’s cladding layer is mostly equiaxed crystals with fine grains, and the surface structure of Sample B’s cladding layer is mostly composed of elongated dendrites.

(2) The microhardness of Sample B is higher than that of Sample A. This is because there are more bcc phases in Sample B. In the bcc phase, the lattice resistance of dislocation movement is larger, and the ability to resist plastic deformation is enhanced; however, the alloying elements on the surface of Sample A volatilize and burn out under the action of high-energy-density laser beams. It makes the surface structure appear to have fine cavities, causing a decrease in hardness.

(3) The abrasion resistance of Sample B is about 2.7 times higher than that of Sample A. The wear mechanism of Sample A is adhesive and abrasive wear, and for Sample B it is abrasive wear. This is due to the loose surface structure and concentrated hard phase of Sample A. In the process of continuous friction, large flaking and deep furrows are produced. The surface layer structure of Sample B has good bonding strength, forming a good wear-resistant framework perpendicular to the growth direction of the thin rod dendrites and improving its wear resistance.

## Figures and Tables

**Figure 1 materials-14-02839-f001:**
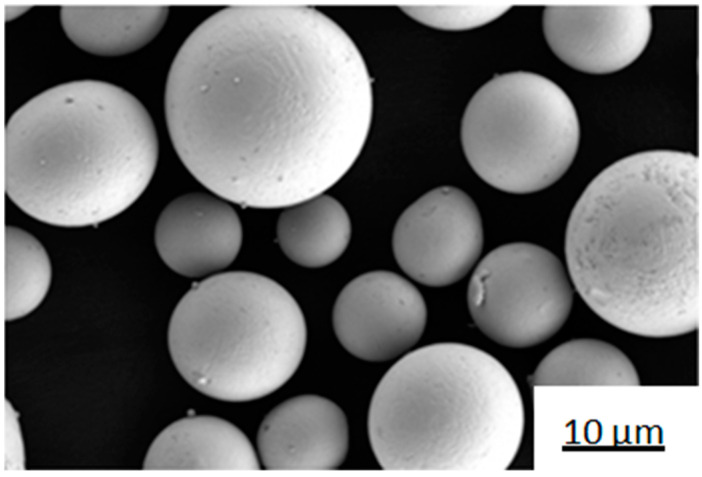
Micromorphology of Fe-based alloy powder.

**Figure 2 materials-14-02839-f002:**
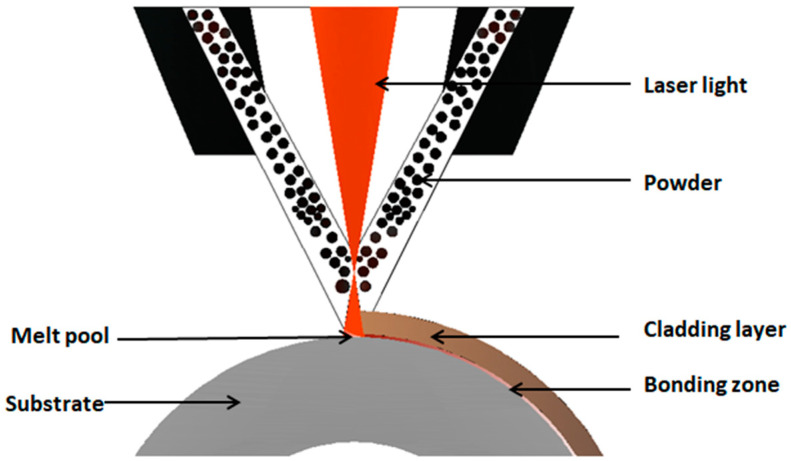
Schematic diagram of the laser cladding process.

**Figure 3 materials-14-02839-f003:**
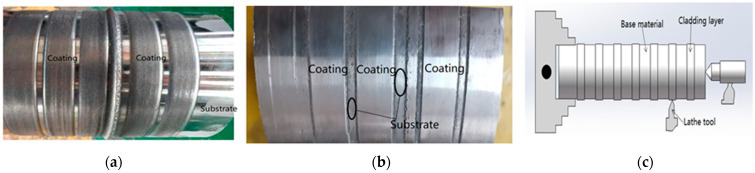
Molten pool morphology and turning diagrams after laser cladding and turning treatment. (**a**) Melt pool morphology; (**b**) shape after turning; (**c**) schematic diagram of turning to processing.

**Figure 4 materials-14-02839-f004:**
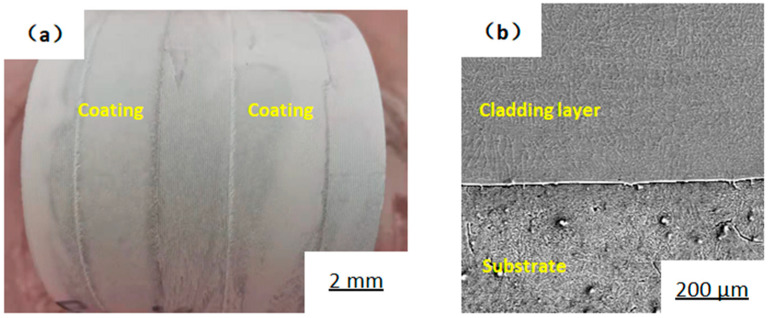
Flaw detection of cladding layer and its cross-section metallographic microscope observation. (**a**) Flaw detection; (**b**) microscopic observation of cladding layer.

**Figure 5 materials-14-02839-f005:**
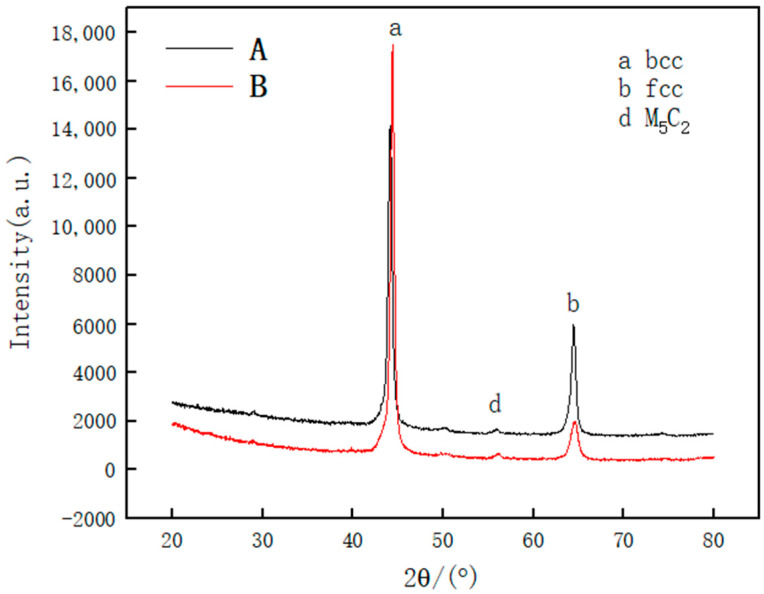
XRD spectra of cladding layers of Samples A and B.

**Figure 6 materials-14-02839-f006:**
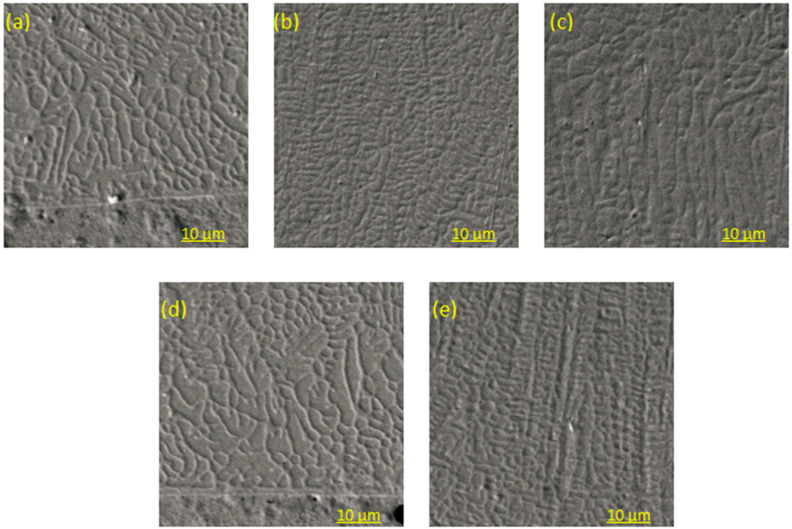
Microstructure of the cladding layers of Samples A and B. Organization morphology of Sample A: (**a**) lower area of the cladding layer; (**b**) middle area of the cladding layer; (**c**) upper area of the cladding layer. Organization morphology of Sample B: (**d**) lower area of the cladding layer; (**e**) middle area of the cladding layer.

**Figure 7 materials-14-02839-f007:**
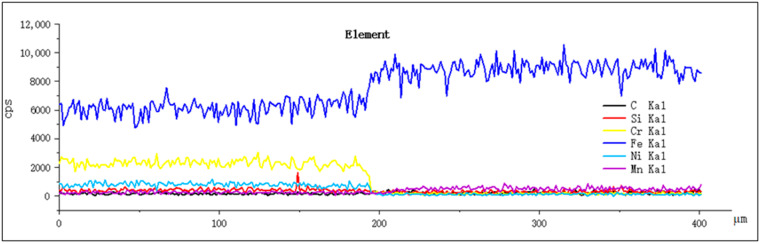
Diffusion diagram of elements at the interface between the cladding layer and the substrate.

**Figure 8 materials-14-02839-f008:**
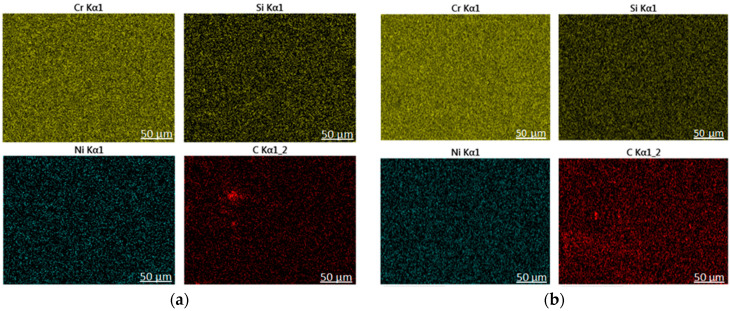
Scanning of the upper and middle regions of the cladding layer. (**a**) Upper area of the cladding layer; (**b**) middle area of the cladding layer.

**Figure 9 materials-14-02839-f009:**
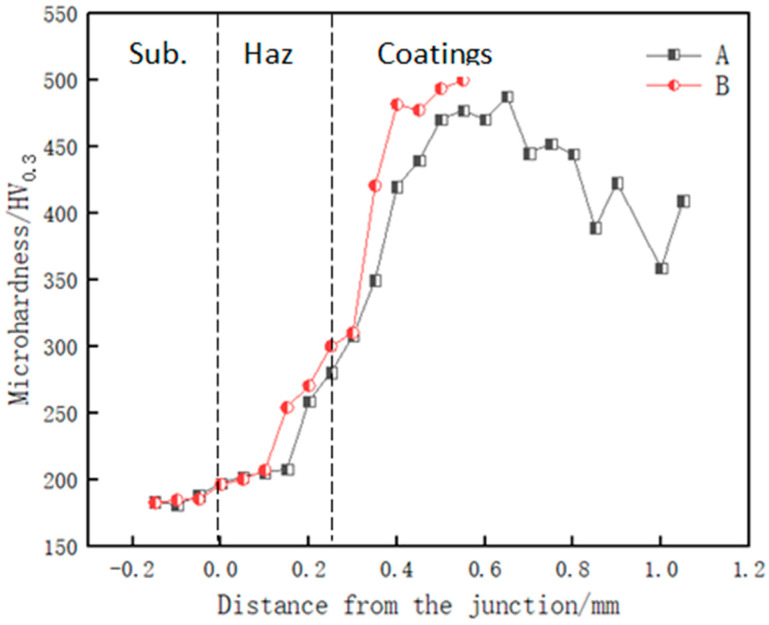
Microhardness distribution of cladding layers of Samples A and B.

**Figure 10 materials-14-02839-f010:**
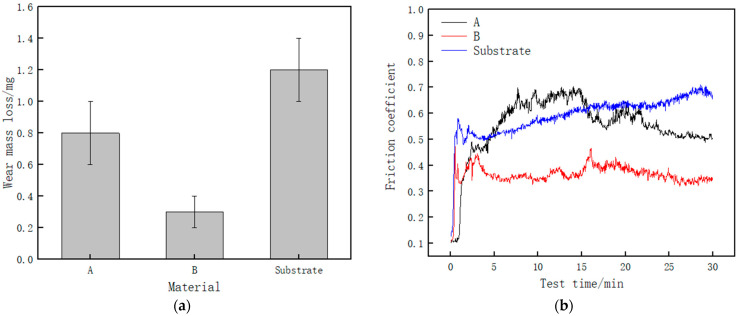
Curves of wear and friction coefficient of cladding layer and substrate. (**a**) Wear of cladding layer and substrate; (**b**) curve of the friction coefficient between cladding layer and substrate.

**Figure 11 materials-14-02839-f011:**
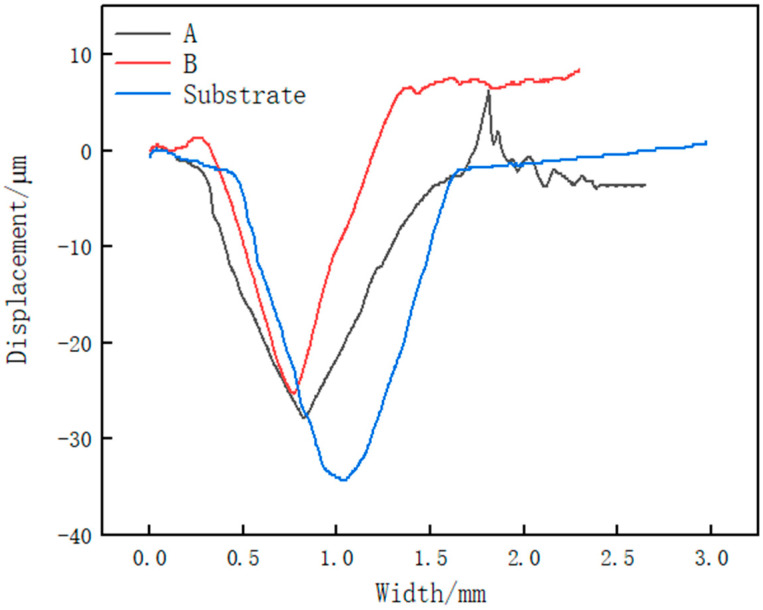
Wear profile of cladding layer and substrate.

**Figure 12 materials-14-02839-f012:**
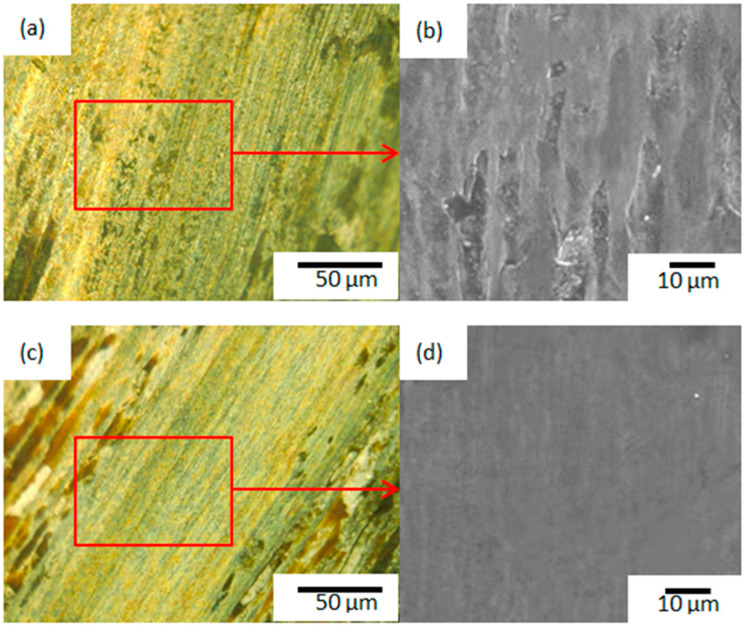
Wear morphology of Samples A and B. (**a**,**b**) The overall morphology and local magnification of the wear scar of Sample A; (**c**,**d**) the overall morphology and local magnification of the wear scar of Sample B.

**Table 1 materials-14-02839-t001:** Chemical composition (mass fraction, %) of laser cladding powder and matrix (27SiMn steel, wt%).

Samples	C	Cr	Si	Mn	Mo	P	Ni	B	S	Cu	Fe
27SiMn	0.24–0.32	≤0.30	1.1–1.4	1.1–1.4	—	≤0.04	≤0.30	—	≤0.04	≤0.30	Bal
Alloy powder	0.1	17.4	1.15	0.28	0.3	—	2.85	1.00	—	—	Bal

**Table 2 materials-14-02839-t002:** Friction and wear test parameters.

Load (N)	Length (mm)	Reciprocating Motor Speed (r min^−1^)	Friction Speed (m min^−1^)	Friction Time (min)	Friction Stroke (m)
30	5	500	5	30	150

## Data Availability

Not applicable.

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
