# Peer review of "Microstructure and Wear Resistance of Laser Cladding of Fe-Based Alloy Coatings in Different Areas of Cladding Layer"

_materials, 2021, doi:10.3390/ma14112839_

Round 1
Reviewer 1 Report
The work of Q. Bai et al is focused on the development of additive laser technology. Currently, this technology is very relevant for the formation of wear-resistant coatings. The most interesting results of the present work is the dependence of microhardness and wear resistance on the depth of the cladding layer. The text of the article is well structured, the conclusions are confirmed by experimental results. The work is recommended for publication in the journal Materials after clarifying some points and text corrections:
- What the laser was using for the realization of laser cladding (wavelength, mode)?
- It is necessary at the first mention in the text to give a definition for “BCC phase” and “FCC phase”.
- The average particles size of the Fe-based alloy powder in the text is given in units of "mesh", and in Figure 1 - "μm". It is necessary to convert to one unit.
- The authors associate the change in microhardness along the depth of the coating with carbides formation, but in Figure 5 the peaks of carbides are the same in the upper and middle parts. Perhaps the formation of the FCC phase has a greater influence here?
- According to Figure 7, the Mn content in the cladding layer is higher than in the substrate, which contradicts the data in Table 1.
- Lines 243 and 244: “HV” should be replaced with “±”.
- Line 277: “Figure 10(b) shows the friction coefficient curve between the cladding layer and the substrate.” What does it mean?
- Lines 196 and 272. It is necessary to remove the coloring of the text.
- In Figure 12 it is necessary to check the given scales.
Author Response
Dear reviewer:
Thank you for your comment on our manuscript entitled " Microstructure and wear resistance of laser cladding Fe-based alloy coatings in different cladding areas, ID: materials-1221556. These comments are valuable, very helpful to revise and perfect our paper, and have important guiding significance for our research. We have carefully studied the comments and made corrections, hoping to get approval. The revised part is marked in red on the manuscript. The main corrections in this article and the responses to the reviewer’s comments can be found in the word file: Please see the attachment.

Reviewer 2 Report
The authors have presented research of potential interest to many readers involved in many aspects of hydraulic machinery. Properties of Fe-based alloy coating obtained using laser cladding were investigated, including morphology, microstructure analysis, hardness measurement, abrasion resistance, and wear analysis.
However, the article has some serious flaws. There is no proper conclusion. English style is extremely poor - in some cases, the article sounds like an instruction manual (77-81, 101-104). This poor language style and grammar errors significantly decrease the quality of the presentation - which, in its essence, has scientific soundness.
If this manuscript should be considered for publication, extensive editing of the English language is required, together with the proper conclusion which emphasizes the scientific contribution of the research.
Reviewer 3 Report
The paper presents research on the study of microstructure, phase composition, hardness and wear resistance of Fe-based alloy coating on 27SiMn substrate; the cladding layer was obtained by laser cladding technology. The obtained results are clearly presented and discussed with the results of other authors. Therefore, I have only a few small comments on the article, mostly of a formal nature.
- please check the names and surnames of the authors, is there a space between the first and last name?
- line 91 -107 : From the text in Chapter 2 .2. I did not understand when the process parameters no. b, c were used. You state "The selected and optimized set of process parameters is set a: .......
"The cladding test was carried out with this combination of process parameters, and then the surface of the obtained cladding layer was treated with a lathe ". Do you mean a combination of a, b, c processes? - explain the abbreviations fcc, bcc.
- Figure 5: Why isn't the scale on the y-axis?
- References are not in accordance with the guidelines for authors.
Reviewer 4 Report
The manuscript submitted for review concerns the study of the microstructure and wear resistance of Fe-based coatings produced by the laser cladding method. The article is quite chaotic. I have some comments. Perhaps it will help to correct the manuscript:
- The executive summary clearly outlines what the manuscript is about. I believe that it is good and does not require major changes.
- Why do the authors rate additive technologies as green (line 38)? Is it the authors' judgment or is it generally accepted by many researchers?
- The authors take up a practical topic that deserves a good word. They create a coating on a product that is used in industry.
- I believe that the level of the English is not satisfactory. I think that contact with an English teacher or nativ speaker will significantly improve the language quality. It's not about perfect English, but in my opinion it should be improved.
- Figure 2. Enlarge the descriptions in the diagram
- Line 105. There is no space after "Table 2"
- Line 110. Did the authors mean EDM machine?
- The journal requires the origin of the research equipment in brackets. Adapt the manuscript to the journal's requirements.
- Line 120. 300 g is hardness. The microhardness is in the load range of 1-200 g. Some standards state up to 500, so I don't consider it a mistake. But perhaps it is worth referring to the norm.
- Figure 4 is of fairly poor quality.
- Figure 5. Peak "c" is difficult to identify.
- Please explain exactly what is the reason for the better wear resistance of the middle area of ​​the coating. Did both samples submitted for testing have the same initial surface roughness? They should have, because only then can the consumption of samples be compared.
- Please write conclusions presenting the novelty of this work, not a description of the research results.
- Pay attention to the correctness in both the content and the references. There are a lot of typos.
Round 2
Reviewer 2 Report
The authors have responded to the suggestions in the review, and have made the appropriate improvements in the manuscript.
I believe that the manuscript can be accepted in the present form.
Reviewer 4 Report
In my opinion article should be published in present form.